# Equilibrium and Dynamic Surface Tension Behavior in Colloidal Unimolecular Polymers (CUP)

**DOI:** 10.3390/polym14112302

**Published:** 2022-06-06

**Authors:** Ashish Zore, Peng Geng, Michael R. Van De Mark

**Affiliations:** Department of Chemistry, Missouri S&T Coatings Institute, Missouri University of Science and Technology, Rolla, MO 65401, USA; aszbnd@umsystem.edu (A.Z.); pgkr4@umsystem.edu (P.G.)

**Keywords:** colloidal unimolecular polymer (CUP), single-chain polymer nanoparticle, surface tension, maximum bubble pressure tensiometer (MBPT), counterion condensation, diffusion coefficient

## Abstract

Studies of the interfacial behavior of pure aqueous nanoparticles have been limited due tothe difficulty of making contaminant-free nanoparticles while also providing narrow size distribution. Colloidal unimolecular polymers (CUPs) are a new type of single-chain nanoparticle with a particle size ranging from 3 to 9 nm, which can be produced free of surfactants and volatile organic contents (VOCs). CUP particles of different sizes and surface charges were made. The surface tension behavior of these CUP particles in water was studied using a maximum bubble pressure tensiometer. The equilibrium surface tension decreased with increasing concentration and the number of charges present on the surface of the CUP particles influences the magnitude of the interfacial behavior. The effect of electrostatic repulsion between the particles on the surface tension was related. At higher concentrations, surface charge condensation started to dominate the surface tension behavior. The dynamic surface tension of CUP particles shows the influence of the diffusion of the particles to the interface on the relaxation time. The relaxation time of the CUP polymer was 0.401 s, which is closer to the diffusion-based relaxation time of 0.133s for SDS (sodium dodecyl sulfate).

## 1. Introduction

Surface tension is a crucial property that has significance in many industries, including the fields of coatings, adhesives, inks, etc. The growing use of nanoparticles and colloidal suspensions in these industries also generates interest in understanding the contribution to surface tension behavior made by these charge-stabilized particles in the absence of any surface-active ingredients [1,2,3,4,5]. Studies have shown the use of finely spread particles at the oil–water interface to stabilize foams and emulsions [6,7,8] However, making charge-stabilized colloidal suspensions free of surface-active ingredients or any contaminants has been difficult and often involves time-consuming and complicated processes such as dialysis, ultrafiltration cells and ion-exchange resin, etc. [9]. This purification issue makes studying the surface tension behavior of nanoparticles difficult as the presence of trace amounts of impurities can affect or dominate the measurements.

Colloidal unimolecular polymer or CUP particles are typically 3–9 nm in size, charge-stabilized particles that are simple and easy to prepare [10]. These CUP particles are made from a single polymer chain, with a well-balanced number of hydrophobic and hydrophilic units, which collapses into a particle by a simple process called water reduction (Figure 1). The polymer chains collapse into a particle because the polymer–polymer interactions become stronger than the polymer–solvent interactions, similar to the formation of micelles. The charge groups repel each other, pushing them apart, which causes the chains to conform into a spheroid during the collapse. The charges will try to distribute themselves evenly on the particle surface to minimize the charge–charge repulsion forces. The charged groups present on the surface of the particle prevent aggregation by providing stability through ionic repulsion. The water reduction process gives stable colloidal dispersion that is free of additives, surfactants, volatile organic compounds (VOCs) or any form of impurities. The CUP suspension thus prepared contains only charged particles, water and counterions, along with a relatively small amount of the base to keep the pH (8.5–9.0) consistent. It is easy to manipulate the physical parameters, such as particle size and charge density, of the CUP surface and the polymer composition of these CUP particles [11]. The spheroidal conformation of the CUP particles has been confirmed using AFM (atomic force microscopy) imaging [12]. However, CUP particles have a strong tendency to form clusters or aggregates when they dry. This makes observing isolated particles and getting a good size distribution difficult when using image analysis techniques such as AFM or TEM (transmission electron microscopy). CUP particles can offer good model material for studying proteins; they can also have potential applications in the fields of coatings, drug delivery, the catalyst matrix and many other areas. These CUP particles have a layer of surface or bound water, with many different properties such as density, specific heat capacity, freezing point, NMR relaxation time, etc., compared to regular/bulk water [13,14,15]. The charges present on the particles and the surface water also gives rise to electroviscous effects [16].

In the field of coatings, CUPs can be used both as a coating resin and in conjunction with latex and polyurethane dispersions (PUDs), wherein they can also be cured with an aziridine [17] or a melamine [18] crosslinker. CUP particles can be made using sulfonic acids as the charge-stabilizing group [19], which can be used as a catalyst for waterborne acrylic-melamine systems [20]. CUP particles with cationic charged groups have been made using QUAT monomer ((2-(methacryloyloxy) ethyl) trimethylammonium chloride) [21] or amines with acetic acid to generate the cationic salt [22]. CUPs with amine functionally charged groups have also been used as a crosslinker for waterborne epoxy coatings. CUP particles have also proven to be a useful additive for freeze-thaw stability and wet-edge retention, due to the presence of non-freezable water around them [22].

Surface tension is one of the most important properties in coatings and is controlled primarily using surface-active agents. CUP particles can alter the surface tension of water and it is, therefore, important to understand the interfacial behavior of these particles at the air–water interface. Surface tension studies [23,24,25] conducted with polyelectrolyte solutions show that surface activity is due to the orientation of hydrophobic and hydrophilic groups in the polymer chain at the air–water interface. The polyelectrolytes are present in the solution in a free-moving open-chain configuration that makes possible the orientation of hydrophobic groups at the interface. For a CUP particle, the polyelectrolyte chain is collapsed, such that the hydrophobic groups are mainly present in the interior of the particle. The behavior of the CUP particle at the air–water interface is similar to that of solid-charge-stabilized colloidal particles, rather than freely moving or flexible polyelectrolyte chains. When the glass transition temperature of the CUP polymer is high (above ambient temperature), the hydrophobic groups present in the interior are not mobile. Studies have been performed to understand the surface behavior of small charge-stabilized particles, such as silica [26,27], TiO_2_ [28] and polystyrene [9] at the air–water interface. A theoretical model was developed by Paunov [29] for understanding the adsorption of charged colloid particles at the air–water interface. The studies conducted by the authors of [26,28] with TiO_2-_ and SiO_2_-based charge-stabilized colloidal particles used large-sized particles (with an average size much greater than 30 nm) with a very broad size distribution, and they contained contaminants or supernatants that were present in the dispersion. Surface tension studies of charge-stabilized particles of a size less than 10 nm have rarely been reported. One of the difficulties has been to make a charge-stabilized nanoparticle that is free from any other ingredients. The nanoscale dispersions of inorganic particles such as bismuth telluride [30], aluminum oxide and boron nanoparticles [31] have been successfully studied to gain insight into their surface tension behavior. These studies attributed the decrease in surface energy to the electrostatic repulsion between the particles. A preliminary study on equilibrium and dynamic surface tension was conducted using CUPs with carboxylate, sulfonate and QUAT-based ionized groups [32]. Sulfonates showed lower surface tension compared to QUATs, followed by carboxylates. The surface tension behavior of CUP particles was also compared against polyurethane dispersions (PUDs) and latex. Latex and PUDs, due to their large particle size, have slow diffusion and, therefore, take longer to reach equilibrium.

The study presented here focuses on both equilibrium and the dynamic surface tension behavior of CUP particles at the N_2_–water interface. Air contains 78% N_2_; therefore, using pure N_2_ helps us to understand air interface behavior without risking any carbon dioxide contamination. The effect of concentration, polymer structure, particle size and charge density (ions per nm^2^) on interfacial behavior was of primary interest in this evaluation. A recent investigation on the effect of CUP particles on the evaporation rate of water provided an important insight into particle arrangement at the interface [33]. Since evaporation and surface tension are both interfacial phenomena, this investigation incorporates both studies to better understand the interfacial behavior of CUP particles. In dynamic surface tension, the bubble rate is varied from fast to slow to create a new surface of a different surface age. When a new surface is created, the CUP particles migrate to the new interface; dynamic surface tension can provide information about the mechanism and the diffusion behavior of particles. Such a dynamic interfacial study can be more useful in practical applications such as spraying, printing, foaming, or coating, which occur under non-equilibrium or dynamic conditions. The maximum bubble pressure method used in this study allows the measurement of both dynamic and equilibrium surface tensions, without the effects of humidity, air turbulence, and contamination by carbon dioxide [34,35,36].

## 2. Materials and Methods

### 2.1. Materials and Synthesis

The purification of chemicals, the polymerization procedure and the water reduction process to form CUP particles for polymers 1–8 are reported elsewhere [13]. The characterization of polymer (molecular weight, polydispersity index, acid number and dry polymer density) and particle size measurements (DLS) of CUP particles were performed using an instrumentation procedure described in detail by the authors of [13]. The molar quantities of monomers—methyl methacrylate (MMA) and methacrylic acid (MAA)—the initiator (Azobisisobutyronitrile, AIBN), chain transfer agent (1-dodecanethiol) and solvent (Tetrahydrofuran, THF) for making polymers 1–8 are mentioned in Table 1. Heptanoic acid and octanoic acid were purchased from Sigma Aldrich, Burlington, MA, USA and used as received. Sodium heptanoate and sodium octanoate were prepared by mixing equimolar quantities of the carboxylic acid with sodium hydroxide (0.1M solution). For the surface tension measurements, solutions of sodium heptanoate and sodium octanoate were prepared in deionized water. 

### 2.2. Surface Tension Measurements

The SensaDyne PC-500 LV, a maximum bubble pressure method (MBPM)-based instrument, was used to measure the surface tension of the CUP suspensions. A constant-temperature water bath was used to equilibrate the temperature of the suspension at 25 ± 0.1 °C before taking the equilibrium surface tension measurements, and at 22 ± 0.1 °C for dynamic surface tension. The tensiometer was calibrated with an analytical reagent of 100% absolute isopropyl alcohol and Milli-Q ultrapure water. The flow pressure of the nitrogen gas was maintained at 40 psi. An average of three readings with less than 0.1 mN/m difference was reported. The surface age used for measuring the equilibrium surface tension was 3 s. 

### 2.3. Thermogravimetric Analysis Measurements 

Thermogravimetric analysis at atmospheric pressure was performed on a TA Instruments Q500 device (TA Instruments, New Castle, DE, USA). Nitrogen was used as the inert gas at a constant flow rate of 40 mL/min. A sample amount of approximately 30 µL was loaded onto tared platinum pans via a micro-pipette to maintain the same depth of the solution. The platinum pans, also sourced from TA Instruments, had a diameter of 9.4 mm. To minimize evaporation before reaching the correct temperature, the sample was heated to the experimental temperature of 298.15 K at 100 K/min. The instrument has a built-in thermocouple placed inside the pan for measuring the temperature of the sample. The sample was held isothermally at 298.15 K for 360 min and the weight percentage change of the sample was recorded as a function of time. Each CUP solution test was run three times.

## 3. Results and Discussion

### 3.1. Polymer Synthesis and Characterization

Polymers were created, such that they had different molecular weights, different monomer ratios of the hydrophobic (MMA) and hydrophilic (MAA) monomer, and a different number of charges per unit area on the surface charge density of the CUP particle. Polymers 1–3 have the same monomer ratio but have different molecular weights, which gives them different charge densities. Polymers 2, 4 and 5 have the same charge density but have different molecular weights. All polymers except for polymers 2, 4, and 5 show a variation in charge density. Variations in molecular weight will result in CUP particles of different diameters. The relationship between particle size and molecular weight will be discussed later. Note that all molecular weights for the polymers are the absolute number average since the real molecular weights define the collapsed size, whereas the relative molecular weights would not do so. Table 2 shows the acid number, density and molecular weight of the copolymers used for this study. The molecular weight and density of the dry CUPs were used for calculating the particle size. 

### 3.2. Particle Size Analysis and Charge Density

Table 3 shows the measured particle size for the copolymers and the calculated particle size from the absolute molecular weight from GPC data. The diameter of the CUP particles was calculated from their molecular weight, using Equation (1):(1)$$d=\sqrt[{3}]{\frac{6M_{w}}{\pi N_{A}\rho_{p}}}$$
where *d* is the diameter of the particle, *M_W_* is the number of the average molecular weight of the CUPs, *N_A_* is the Avogadro number and *ρ_p_* is the density of the dry polymer. As expected, the diameter of the CUP particle increases with an increase in molecular weight, which was consistent with the findings of our previous work [11]. For a unimolecular collapse into a spheroidal conformation, the measured size from DLS should be very close to the calculated size of the molecular weight, as shown in Table 3.

Charge density is the number of charges present per unit area (nm^2^) of the particle, and is calculated using Equation (2):(2)$$\rho_{v}=\frac{M_{W}}{4\pi r^{2}\left(n\times M_{H1}+m\times M_{H2}+\ldots \ldots \ldots +M_{i}\right)}$$
where *n* and *m* are the statistical number of hydrophobic monomers 1 and 2 in a repeat unit, also mentioned as the monomer ratio, *M_W_* is the molecular weight of the CUP, *M_H_*_1_ and *M_H_*_2_ are the molecular weights of hydrophobic monomers 1 and 2, *M*_i_ is the molecular weight of the hydrophilic monomer and *r* is the radius of the CUP particle. The charge density of the CUP particle can easily be manipulated by changing the molecular weight of the polymer/particle size and/or the composition (monomer ratio) of the polymer. The dumbbell conformation of polymer 8 will be discussed later, in Section 3.3. The charge density for the dumbbell was calculated from Equation (2) using ‘8*πr*^2^’ instead of ‘4*πr*^2^’, where *r* is calculated from Equation (1) using ‘*M_w_*/2’ instead of ‘*M_w_*’. The two halves of the polymer chain form the two spheres of a dumbbell shape.

### 3.3. Equilibrium Surface Tension Behavior 

The bubble tensiometer required a bubble rate slow enough to allow equilibrium to be established. The surface age of three seconds that was chosen was long enough to allow the CUP particles to reach equilibrium at the interface. The equilibrium surface tension of all the CUPs that were measured show linear decreases with increasing concentration, followed by curves, and finally become constant at high concentrations, as seen in Figure 2. This behavior of a reduction in surface tension with increasing concentration was also observed for typical surfactants [37]. When comparing CUPs (see Table 4) against an ionic surfactant, such as SDS, at the same concentration (0.001 M), all the CUP polymers showed a smaller reduction in surface tension (Δγ) than SDS [37,38]. Polymer 3 CUPs show the largest difference of Δγ = 4.2. The slope of Δγ/Δc better illustrates the effectiveness of the surface-active agent in reducing the surface tension. The surface tension value of QUAT [21] and sulfonate [19] CUPs of molecular weight 55K and 56K and charge density 0.52 and 0.58 ions/nm^2^, respectively, are shown in Table 4. The Δγ/Δc values of carboxylate CUPs (polymer 3) were closer to those of QUAT CUPs. The higher effectiveness of sulfonate CUPs compared to carboxylates can be attributed to the strong electrostatic repulsion of the sulfonate groups. A comparison study conducted using sulfonate and carboxylate ionomers showed stronger ionic interaction in sulfonates, which was attributed to greater polarization [39]. Sodium salts of carboxylic acid (see Table 4) have also been known to show some surface activity in water [40,41]. Sodium formate shows an increase in surface tension with a concentration similar to that of NaCl, which could be attributed to the absence of hydrophobic groups. In the case of sodium acetate and sodium benzoate, they show a surface activity much like surfactants but at a much higher concentration. Sodium laurate (at pH = 8.5), however, shows much higher surface activity. The size of the hydrophobic group affects surface activity, as seen from the Δγ/Δc values of sodium acetate, sodium benzoate and sodium laurate. For the same concentration, the Δγ/Δc value of CUPs had a larger effect than sodium acetate and benzoate but had less effect than sodium laurate. Unlike most surface-active agents, the molar concentration of CUPs may not have a simple relationship. The hydrophobic groups in the CUP particles are not free to move around or orient their chains at the interface, as in carboxylate-based small molecules. The hydrophobic regions in CUPs are larger than the methyl/phenyl group of the carboxylates and are dominated by the ester groups and, most likely, some of the methyl groups on the backbone.

Okubo used monodispersed polystyrene latex particles with a strongly hydrophobic surface, as well as silica particles that have a hydrophilic surface, to study the surface tension behavior of colloids in deionized water without the addition of any surfactant [9]. In general, there was a decrease in surface tension as the particle volume fraction increased. The particle suspensions were described to be liquid-like or gas-like at low concentrations, due to the suspensions being turbid and milky. The decrease in the surface tension for the liquid-like or gas-like suspension was not very significant. However, at higher concentrations, the surface tension significantly dropped with concentration and the suspension formed a crystal-like structure, in which brilliant iridescent colors, due to Bragg’s diffraction, and glittering single crystals were observed with the naked eye. The CUP suspension was clear at all the concentrations measured in this study. The CUP particles, being in the true nano-scale size of 4–7 nm, cannot scatter visible light; hence, they look clear. At high concentrations, the CUP particles are sufficiently stable and do not aggregate. CUP solutions have been utilized for over 10 years without any size change or stability issues. The volume fraction concentration that was measured for the polystyrene and silica suspensions did not exceed 0.1; the surface tension differed by Δγ = 12 for most polystyrene suspensions and by Δγ = 2 for the silica suspensions. The difference in surface activity for polystyrene and silica is attributed to the high hydrophobicity of the surface of polystyrene. Another study conducted by Dong and Johnson [26,28] shows the surface activity of TiO_2_- and SiO_2_-based charge-stabilized colloidal dispersions (pH = 10 and 11, respectively). The surface tension values of TiO_2_ and SiO_2_ decreased with the increase in the concentration of particles. The surface tension dropped to the lowest value at 5% concentration by weight, then reached a plateau for a while before increasing as the concentration increased. The maximum surface tension difference of Δγ = 3.5 for SiO_2_ suspensions and Δγ = 5.2 for TiO_2_ suspensions was observed at a 5% concentration by weight. For CUPs at 5% concentration by weight (Figure 3), a difference of Δγ = 1.9 was observed for polymer 3. However, the maximum difference of Δγ = 5.5 was observed for polymer 3 CUPs at 18% solids, which is higher than that found in SiO_2_ and is close to that of TiO_2_. One of the significant differences between CUPs and TiO_2_ and SiO_2_ particles is the size distribution. The TiO_2_ and SiO_2_ particles used in the study had a very broad particle size distribution, with a size ranging from 40 nm to 1400 nm and 500 nm to 8,000 nm, respectively. CUPs, on the other hand, have consistently shown much narrower particle-size distributions [10,11]. The particle shapes of TiO_2_ and SiO_2_ particles in the suspension were also irregular rather than spheroidal, as in CUPs. Surface tension studies have been conducted with 2.5 nm and 10.4 nm bismuth telluride nanofluids [30], using contact angle measurements on silicon wafers and glass substrates. At 0.0003% concentration by weight, the 2.5 nm suspension showed a difference of Δγ = 26.70, while the 10.4 nm suspension showed a difference of Δγ = 18.67. The surface tension reduction in the case of bismuth telluride was much higher when compared to the CUP, TiO_2_ and SiO_2_ particles. The bismuth telluride particles used in the study were modified using thioglycolic acid, which can interact to form acid dimers at the interface. The number of acid groups on the nanoparticle surface is unknown, which makes it difficult to access the contribution of thioglycolic acid to surface tension reduction as compared to the actual bismuth telluride nanoparticle surface. The pH of the nanoparticle solution is unknown. The acid groups may also cause the particles to adsorb on the silicon and glass interface of the silicon wafer and glass substrates that are used in contact angle measurement. Furthermore, the bismuth telluride nanoparticles used were only stable for a period of from a few hours to a couple of days, whereas the CUP solution, as mentioned earlier, is stable for over 10 years if the pH is maintained at a basic level (~8.5). Studies conducted with 18 nm aluminum oxide [31] and multiwall carbon nanotubes (D = 8−15 nm, L = 10−50 µm), measured using a pendant drop method, only showed an increase in surface tension with the concentration of the particles in water and ethanol. This behavior was different from the bismuth telluride, TiO_2_ and SiO_2_. All the surface tension studies mentioned earlier do not consider one critical aspect, the charge density of the nanoparticle, which may possibly influence the surface tension behavior. This could be due to an inability to precisely manipulate the number of charges on the surface of these nanoparticles to obtain the required charge density.

For CUPs, the effect of molecular weight on surface tension behavior can be understood from the data of polymers 2, 4 and 5 (Figure 2), which have the same charge density but have a different molecular weight. They show a similar reduction in surface tension at the same molar concentration. This indicates a dependency of surface tension on the charge density of the polymer. Polymers 1 and 4 and polymers 2, 6, 7 and 8 (Figure 2), which have similar molecular weights but different surface charge densities, show that higher charge density CUP particles show more of a reduction in surface tension. The data in Figure 2 can be fitted using two lines, the first for the initial decrease and the second for the constant region. The slope of the first fit can be considered as the “effectiveness of the CUP particles” at reducing the surface tension. The more negative the value of the slope, the higher the effectiveness of the CUP particle at reducing the surface tension. The plot of the slope or the effectiveness of CUPs against the charge density is shown in Figure 4. The data follows an exponential trend and later deviates at a very high charge density.

For polymers containing many ionic groups, a theoretical model for the conformation of the chain, based on an electrostatic blob and the scaling theory, was first developed by de Gennes and Pfuety and reviewed by Dobrynin [42]. Depending on the number of charges or ionic groups present on the chain, the conformation can range from an electrostatic blob to a pearl necklace. A theoretical model [43] has been developed for a dilute solution of polyelectrolytes of uniform charge, with the degree of polymerization N, monomer size b, and the fraction of charged monomers f in a poor solvent, having a dielectric of ε. The model predicts the following for a polyelectrolyte of N = 200 monomers: when the polymer chain is uncharged (f = 0), it collapses into a spherical globule; at f = 0.125, the chain collapses into a dumbbell shape and at f = 0.150, the chain collapses into a pearl necklace with three beads. In the case of polymer 8 (f = 0.17), the charge density of the polymer is high enough to cause the chain to collapse into a different conformation instead of a spheroid. The deviation in the surface tension behavior at a higher charge density can be attributed to the change in conformation of the particle from a spheroidal to a non-spheroidal shape. All the other polymers fall into the spheroidal charge density region, where f is between 0.05 and 0.128.

### 3.4. Model for CUP Particles at Interface

A better understanding of the mechanism of the reduction in surface tension caused by the CUP particles requires a model of these particles to be arranged at the N_2_–water interface. In a study on the evaporation rate of water for these CUP solutions, a model of CUP particles arranged at the N_2_–water interface was presented [33]. Since both evaporation rate and surface tension are interfacial phenomena, this model should explain all the results. In a dilute solution at equilibrium, the particles are randomly distributed and stabilized by a combination of Brownian motion, solvation by water and electrostatic repulsion, due to the presence of an electrical double layer around the particles. The particles present in the water phase are constantly experiencing charge-repulsive force from all directions as they are surrounded by other particles. However, the particles at the interface do not have any charge force exerted on them from the N_2_ side. Thus, the other particles around them push each particle toward the N_2_ interface and force it partially out of the N_2_–water interface, as shown in Figure 5a. 

At the interface, the particles can exist in three different states, as shown in Figure 5b (A: CUP particle with a layer of surface water, B: CUP particle with a surface water layer, followed by a layer of N_2_–interface water, and C: CUP particle with no water). The CUP particle surface is highly hydrophilic and has a layer of strongly associated surface water. Hence, model C is the least likely to exist. The particles at the N_2_ interface are very likely to exist, as shown in models A or B. The results from the evaporation rate study are in good agreement with models A or B [33]. When the evaporation rates of dilute CUP solutions were measured, they showed an increase in the evaporation rate of water from the solution over pure water. This increase in the evaporation rate has been attributed to the increase in the surface area caused by the particles when they deform the interface, as shown in models A and B. If model C were to exist, a decrease in the evaporation rate would be expected as that reduces the surface area at the interface. Studies have shown that hydronium and hydroxide ions accumulate at the air–water interface [44,45]. These accumulated ions can exert an electrostatic repulsion force on the CUP particle at the air–water interface. The force exerted by the accumulated interfacial ions is not strong enough to prevent the particle from pushing out of the air–water interface, as confirmed by the increase in the evaporation rate. This is to be expected because hydronium and hydroxide ions are much smaller in size than the CUP particles. Hence, the accumulated interfacial ions must be pushed aside by the approaching CUP particle at the interface. This pushing aside of the accumulated interfacial ions will also result in the slow diffusion of the particle at the interface. As the CUP particles approach the interface, the hydroxide/hydronium ions rapidly re-equilibrate to accommodate the approaching macro-ion. The re-equilibration of the water ions is in the sub-microsecond timescale. Only the hydroxyl ions above the area of the CUP between r_cup_-h and the center may exert a downward force, with the carboxylate on the air/water radius repelling hydroxyl groups, resulting in only a negligible downward force, if any, on the CUP.

### 3.5. Relationship between Surface Tension and Charge Groups, Based on Model A

Figure 6 shows a particle, of radius r_CUP_, that is present at the N_2_–water interface. The CUP particle extends above the N_2_–water interface to a specific height, h.

Since the increase in the evaporation rate is due to an increase in the surface area, the height h for a given concentration can be estimated using Equation (3):(3)$$h=\sqrt{\frac{\Delta R}{R\times \pi }}\times \left(\sqrt[{3}]{\frac{M_{W}}{\rho \times X_{CUP}}}\right)$$
where *h* is the height of the interface water deformation, Δ*R* is the increased evaporation rate compared with water, *R* is the evaporation rate of the CUP solution, *M_w_* is the molecular weight of the CUP, *ρ* is the density of the CUP solution and *X_CUP_* is the weight fraction of the CUP. There are two main assumptions when calculating the height (h) values using Equation (3): (1) the increase in evaporation rate is solely due to an increase in area, thereby neglecting any other effects, if present. (2) The evaporation rate of bulk water is the same as surface water (model A) or N_2_–interfacial water (model B). Surface water has been successfully studied and has been shown to have different properties, such as density, specific heat capacity and a different freezing point to bulk water. Hence, it is possible for the evaporation rates of surface and bulk water to be different as well. However, despite these assumptions, the height values can be crucial for better understanding surface tension behavior. The evaporation rate data for polymers 1–6 at different concentrations were taken from an earlier evaporation rate study [33] and were measured for polymers 7–8. Furthermore, the height values were calculated using Equation (3), while the circumference of CUP at the interface was calculated using Equations (4) and (5):(4)$$r_{c}=\sqrt{r_{cup}{}^{2}-{\left(r_{cup}-h\right)}^{2}}$$
(5)$$circumference=c_{int}=2\pi r_{c}$$

The circumference, *c_int_*, is the length of the interface created by N_2_, water and the CUP particle or surface water. The inverse of charge density (nm^2^/ion) establishes the area occupied by each ion on the surface of the particle. Assuming that each ion occupied a circular area on the surface, the diameter of the charge (*d_c_*) can be calculated. Using the diameter of charge (*d_c_*), we can calculate the number of charges or acid groups present on the circumference (*N_charge_*), using Equation (6):(6)$$N_{charge}=\frac{c_{int}}{d_{c}}$$

Figure 7 shows a plot of the number of charges or acid groups present on the circumference, in terms of the effectiveness of CUPs. As the number of charges or acid groups at the circumference increases, the CUP particles become more effective at reducing the surface tension. The charge on the circumference also explains the trend in charge density, as seen in Figure 4, because the number of charge groups at the circumference is directly related to charge density. Hence, CUPs with a high charge density show lower surface tension.

A plausible explanation for the trend observed in Figure 7 is that the charges or acid groups present on the circumference, which is also the N_2_-water interface, behave as a surfactant. This can be visualized using Figure 8a, which shows a charge group acting as a surfactant, where the charge or acid group is the hydrophilic head and the hydrophobic surface around it is the hydrophobic tail. When there are more acid groups present on the interfacial circumference, this corresponds to having more surfactant molecules at the interface; hence, the surface tension becomes lower. The concentration (c*) of the charge groups present at the interface for the CUP particles at a given concentration, c (mols/L), can be calculated using Equation (7):(7)$$c^{*}=c\times N_{charge}.$$

The *c** values for polymers 2, 3 and 7 are shown in Table 4. The slope values show the effectiveness of CUP particles to be between that of sodium benzoate and sodium heptanoate. However, as shown in the depiction in Figure 8a, the hydrophobic region of the CUP does not form a linear chain similar to sodium heptanoate. Unlike sodium benzoate and heptanoate, the hydrophobic region also comprises ester groups, and it extends not only above the surface but also to the left, to the right and below it. 

### 3.6. Surface Tension at Higher Concentration

The surface tension deviates from linearity at high concentrations and eventually reaches a constant value (see Figure 2). Similar behavior was also observed in surfactants, where due to micelle formation, the surface tension becomes constant [38,47]. Sodium acetate and sodium benzoate also show a constant surface tension at higher concentrations, which could be due to the formation of loose aggregates instead of a proper micelle [39]. In the case of CUP particles, there is no micelle formation. The surface activity of TiO_2_ and SiO_2_ (pH = 10 and 11), as studied by Dong and Johnson at high concentrations (above 5% solids), also showed a constant surface tension. However, as the particle concentration increased further, the surface tension started to increase. The authors explained this behavior by citing the presence of strong capillary forces between the particles at the interface [48,49]. For colloidal particles stabilized by surface charges (ionic), when the concentration of particles becomes very high, the charges present on the surface can undergo intermolecular counterion condensation or Manning condensation, where some of the charges or surface ions will recombine with the counterions. Intermolecular counterion condensation has been observed in CUP solutions and its effect on surface water thickness has been studied in papers on thermodynamic characterization [13] and the electroviscous effect [16]. Due to intermolecular counterion condensation, the number of charges or ionized acid groups present on the surface reduces, thereby reducing its effective charge density. The surface tension results shown in Figure 2 can be fitted using two linear fits to obtain an intersection point. The molar concentration at the intersection can be considered as the onset concentration for intermolecular counterion condensation. The interparticle distance at the onset concentration can be easily estimated using Equation (8):(8)$$Inter-particle\;distance,\;IPD=\frac{1}{\sqrt[{3}]{z}},\;\;z=number\;concentration\;in\;m^{-3}$$

The number concentration, *z*, is the number of particles present in one cubic meter of solution. When the interparticle distance at the onset concentration was plotted against charge density, it indicated linear behavior, as shown in Figure 9. Having a higher charge density will increase the repulsive force between the particles; hence, counterion condensation can be expected at lower concentrations. Low charge-density particles must come closer to each other for counterion condensation to take place.

The surface tension becoming constant at high concentrations can be explained by intermolecular counterion condensation. Counterion condensation reduces the overall number of charges present on the surface, thereby reducing its charge density. This will reduce the number of charges present on the interfacial circumference, as shown in Figure 10. The reduction in the number of interfacial charged groups will cause the surface tension to stay constant, even when more CUP particles are being added to the solution.

### 3.7. Dynamic Surface Tension Behavior

Polymers 1–5 were chosen for analysis, due to their large Δγ in the linear region (Figure 2), which was useful for understanding the effect of concentration by measuring the dynamic surface tension at three different concentrations, along with understanding the effects of size and charge density on the dynamic surface tension. Polymers 6 and 7 do not show a large Δγ in the linear region; hence, they cannot provide a dynamic curve with a good fit. Polymer 8, being in a dumbbell conformation, was not measured for dynamic behavior since it does not fit the spheroidal model. Figure 11 shows the plots of dynamic surface tension for the CUP solutions of polymers 1–5, measured at three different concentrations for each polymer. Surface age, defined as the time interval between the onset of a bubble and the moment of maximum pressure, was manipulated by changing the bubble rate. A slow bubble rate gives a longer surface age, thereby giving more time for the CUP particle to reach the N_2_ (bubble)–water interface. 

The dynamic data for surface tension (*γ*) vs. surface age (*t*) for all the CUP polymers measured show an exponential fit, using Equation (9):(9)γ=γe+Ae−tτ
where *γ_e_* is the equilibrium surface tension and A (amplitude of the exponential curve, *γ_t_*_=0_ − *γ_e_*) and τ (relaxation time) are the fitting parameters. The polymers used for the study were of different molecular weights and charge densities, to explore their effects on dynamic behavior. The relaxation time, *τ*, gives an indication of the rate at which the solution reaches equilibrium; therefore, it gives an idea of the mobility of CUP particles. The relaxation time τ values for all the CUP polymers measured at different concentrations are given in Table 5.

The particle size study using CUPs that was conducted by Van De Mark et al. [10] showed that accurate particle-size measurement of the CUP particles using the DLS technique requires the viscosity of the solvent to be replaced by the viscosity of the solution, to account for the increased viscosity due to the electroviscous effect. The collective diffusion coefficient of the spherical particles can be approximated from the generalized Stokes–Einstein equation, Equation (10), which relates the diffusion coefficient (*D_c_*) to the radius (*r*) of the particle measured using DLS, the viscosity (*η*) of the solution and temperature (*T*).
(10)$$D_{c}=\frac{k_{b}\times T}{6\times \pi \times \eta \times r}$$

As seen from Table 5, the relaxation time shows an increase with an increase in concentration for all the CUP polymers that were measured. This could be due to the lower diffusion coefficient of the particles at higher concentrations. For a given CUP particle, as the concentration increases, it likewise increases the solution viscosity, which has an inverse relation to the diffusion coefficient, as shown by the Stokes–Einstein equation. 

Table 6 shows the relaxation time of CUP polymers 1–5 measured at the same concentration of 1.03 mol/m^3^. There are two variables affecting the diffusion of the CUP particle—particle size and charge density. Charge density gives rise to the electroviscous effect in the solution. Higher charge density leads to strong electroviscous behavior. The diffusion coefficients were calculated, using Equation (10), by measuring the viscosity at 1.03 mol/m^3^ and the particle size on DLS at 22 °C. A plot of the relaxation time against the diffusion coefficient (Figure 12) shows that as the diffusion coefficient increases, the relaxation time decreases. The particles can migrate faster to the newly created N_2_–water interface (bubble). 

Figure 13 shows the dynamic behavior of sodium dodecyl sulfate (SDS) at a 2 mmol concentration [50]. 

The dynamic curve fits a double exponent equation:(11)γ=γe+Ade−tτd+Ake−tτk
where *γ_e_* is the equilibrium surface tension and *A_d_* and *A_k_* (amplitude of the exponential curve, *γ*_*t*=0_ − *γ_e_*) and *τ_d_* and *τ_k_* (relaxation time) are the diffusional and kinetic fitting parameters. In the case of surfactants, the interface adsorption is dependent on diffusion at a short surface age, and on the interfacial organization kinetic at a long surface age. When a new surface is created, the interface is relatively empty and there is no barrier to adsorption at the interface. Hence, the time at this stage (*τ_d_*) is governed by the diffusion rate of surfactant molecules to the interface. When the surface becomes older, the concentration of surfactant molecules at the interface increases, which creates an organizational barrier for the surfactant molecules that are moving to the interface. Hence, the time at this stage (*τ_k_*) is governed by the organization kinetic of the surfactant molecules at the interface. The diffusional and kinetic mechanisms are also observed in other surfactants [51,52]. For SDS, the *τ_d_* and *τ_k_* values are 0.133 and 12.85 s, while the A_d_ and A_k_ values are 3.32 and 3.5 mN/m, respectively. 

All the CUP polymers have shown a single relaxation time (*τ*) at all the measured concentrations. All the τ values are relatively small. Polymer 1, at a 2.1 mmol concentration, showed a τ = 0.401, which is closer to *τ_d_*, compared to the *τ_k_* of SDS at a similar concentration (2 mmol). Therefore, the relaxation times in CUPs are primarily a function of the rate of diffusion. The reestablishment of the charged particles’ distribution as the bubble grows into the solution is relatively rapid and is not dominated by any major structural organizational mechanism. The relaxation time, τ, of all the CUPs (polymers 1–5) measured is higher than the diffusion relaxation time, *τ_d_*, of SDS. This is to be expected, as CUP particles are larger in size compared to the SDS molecule, and CUPs also exhibit an electroviscous behavior when in solution that can further affect the diffusion rate. The presence of the accumulated interfacial hydronium and hydroxide ions may also contribute to the slow diffusion of CUP particles at the interface. However, this should not be a major contributor to the downward force on the CUP particles.

## 4. Conclusions

The maximum bubble pressure tensiometer results regarding CUP particles provide a detailed insight into the equilibrium and dynamic interfacial behavior of pure nanoscale-sized particles. The data from equilibrium surface tension, combined with the evaporation behavior, gave a better model of the particles present at the air–water interface. The model shows that the CUP particles were pushed out of the air–water interface, which caused the surface charges to align at the air–water interface. The surface charges then acted as surfactants, due to the hydrophobic region present around them. The magnitude of the surface tension was closer to those of sodium benzoate and sodium heptanoate, although they are not good models for the behavior of CUP solutions. At higher concentrations, the surface tension became constant, due to surface charge condensation. The surface charge condensation occurred at a longer distance from the surface when the CUP surface charge density was high. The charge condensation reduced the number of charges that acted as surfactants present at the air–water interface. Dynamic surface tension behavior is mainly affected by the diffusion coefficient of the particle, which is dependent on particle size and charge density. Slower particles show a longer relaxation time, indicating that the dynamic behavior is influenced by the rate of diffusion rather than a structure organization mechanism. Unlike surfactants, where the dynamic behavior is a function of diffusional and kinetic mechanisms, CUPs have shown a diffusion-based behavior wherein an organizational mechanism may be either absent or insignificant. 

## Figures and Tables

**Figure 1 polymers-14-02302-f001:**
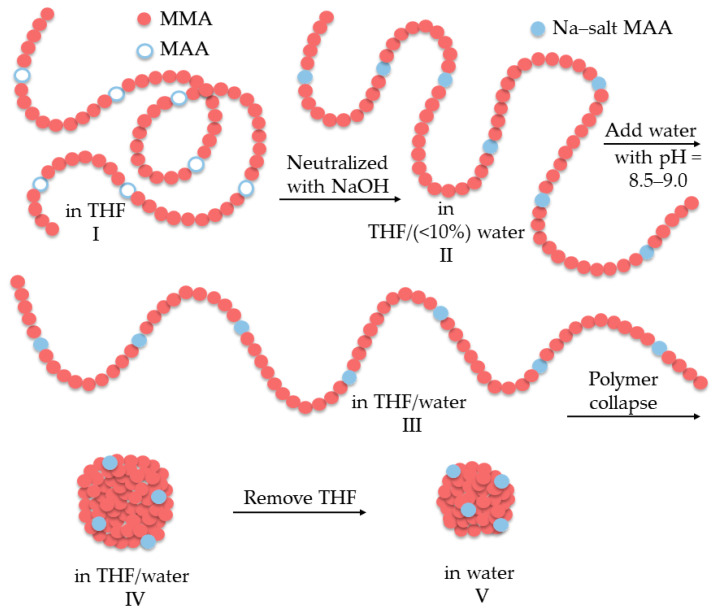
Schematics of the water reduction process and CUP formation [13].

**Figure 2 polymers-14-02302-f002:**
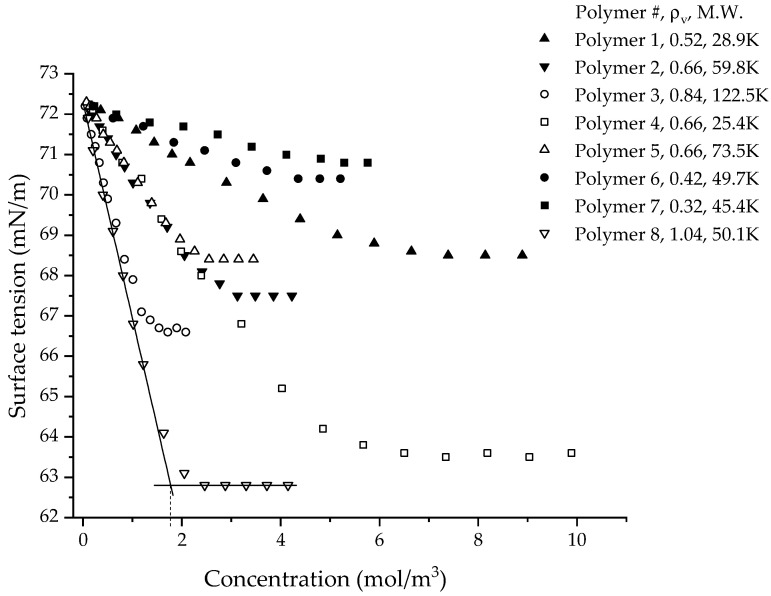
Equilibrium surface tension (mN/m) vs. the molar concentration (mol/m^3^) of CUP solution made from polymers 1–8.

**Figure 3 polymers-14-02302-f003:**
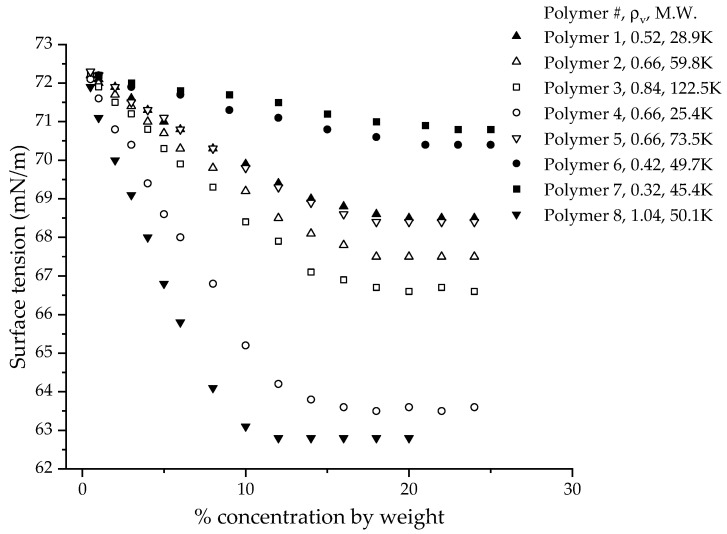
Equilibrium surface tension (mN/m) vs the percentage (%) weight of CUP solution made from polymers 1–8.

**Figure 4 polymers-14-02302-f004:**
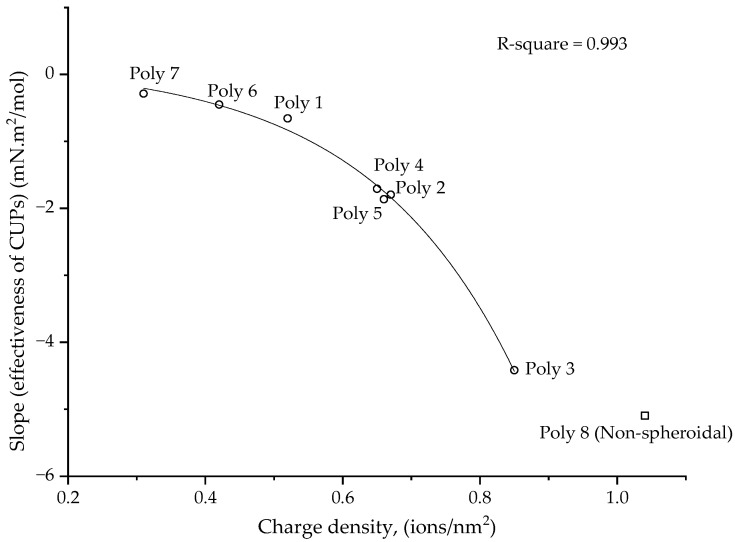
The slope or effectiveness of CUP against the charge density (ions/nm^2^) of the particles.

**Figure 5 polymers-14-02302-f005:**
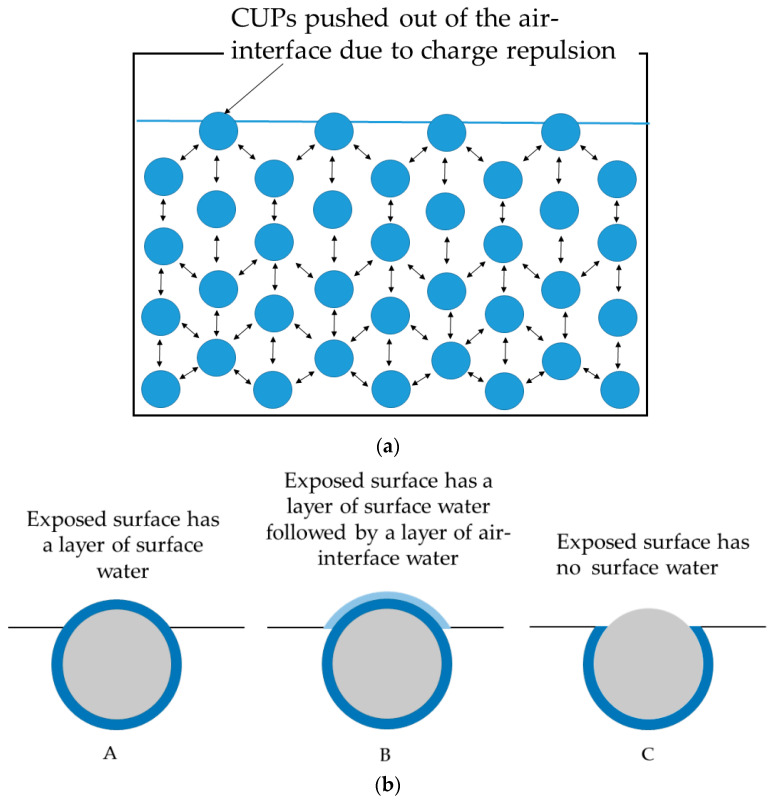
(**a**) CUP particles are pushed through the N_2_ water interface due to charge repulsion from the particles below. (**b**) CUP particles that are pushed through the N_2_–water interface due to charge repulsion can exist in three possible states (A–C).

**Figure 6 polymers-14-02302-f006:**
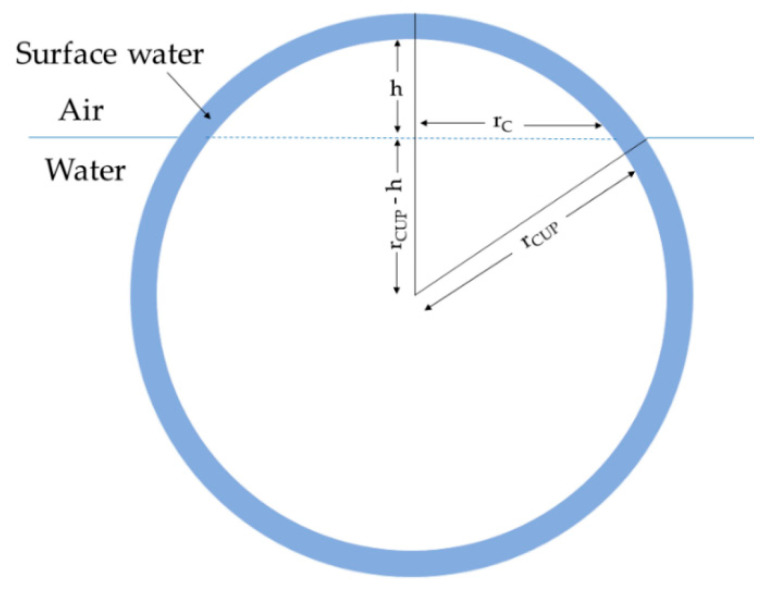
Deformation of the surface by CUP particles at the N_2_–water interface.

**Figure 7 polymers-14-02302-f007:**
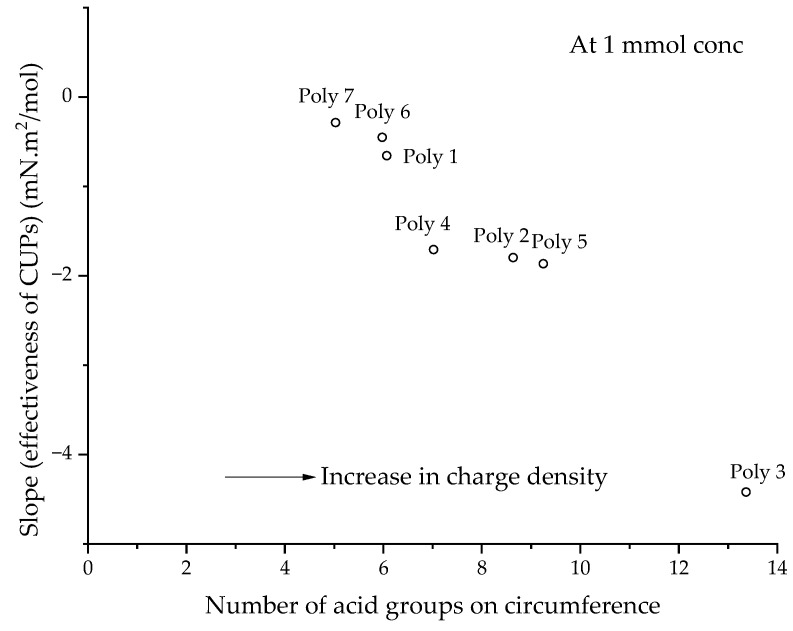
The slope or effectiveness of CUPs against the number of charges or acid groups present on the circumference (*N_charge_*).

**Figure 8 polymers-14-02302-f008:**
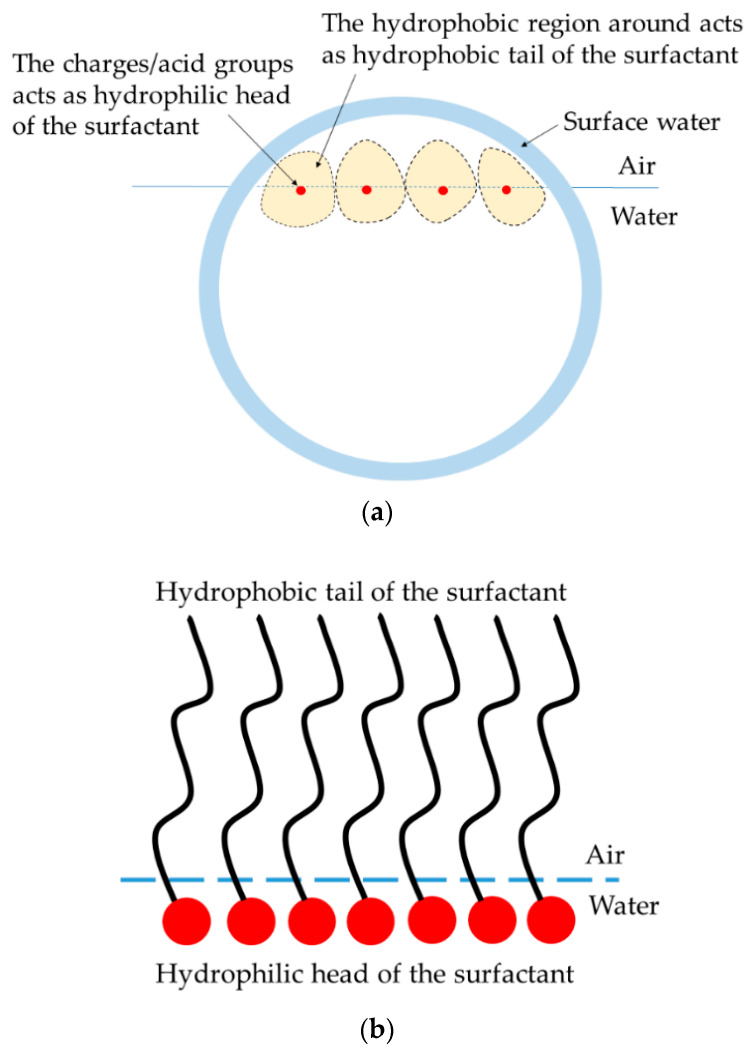
(**a**) Model depiction of charge groups mimicking a classic surfactant at the N_2_–water interface; (**b**) a classic surfactant at the N_2_–water interface [46]. Reprinted (adapted) with permission from Peng, M; Duignan, T.T.; Nguyen, C.V.; Nguyen, A.V. From Surface Tension to Molecular Distribution: Modeling Surfactant Adsorption at the Air–Water Interface. *Langmuir*
**2021**, *37*, 2237–2255. Copyright 2021 American Chemical Society.

**Figure 9 polymers-14-02302-f009:**
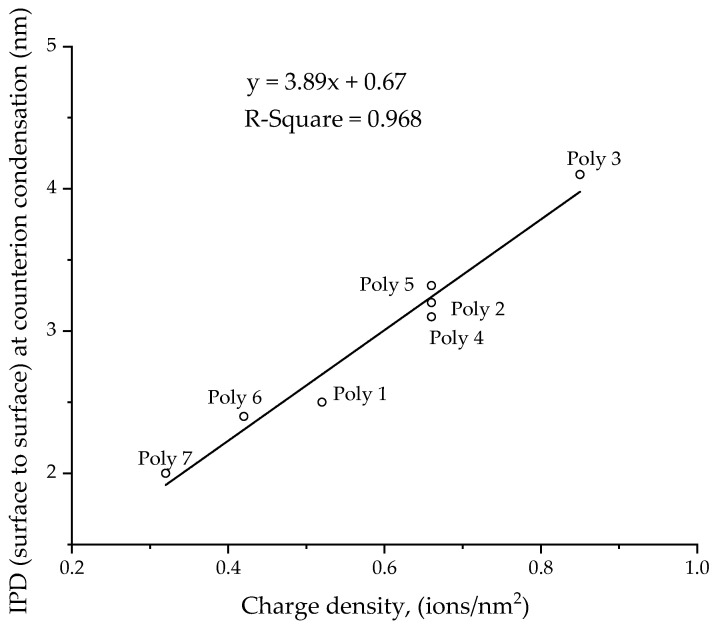
Interparticle distance (nm) at the onset concentration for counterion condensation against the charge density (ions/nm^2^) of the particle.

**Figure 10 polymers-14-02302-f010:**
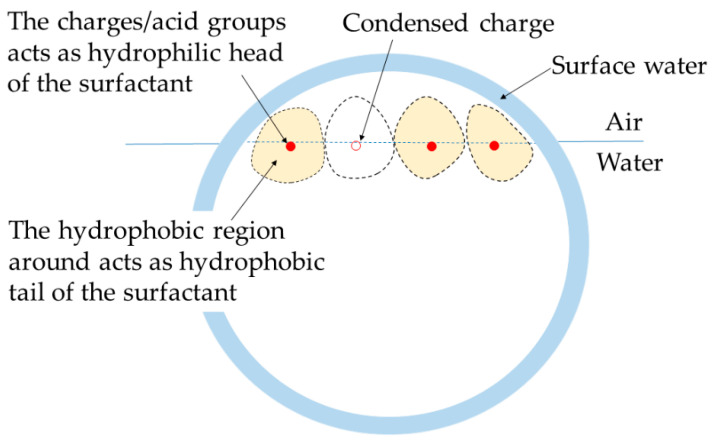
Charge condensation at high concentration, reducing the number of charge groups present at the interfacial circumference.

**Figure 11 polymers-14-02302-f011:**
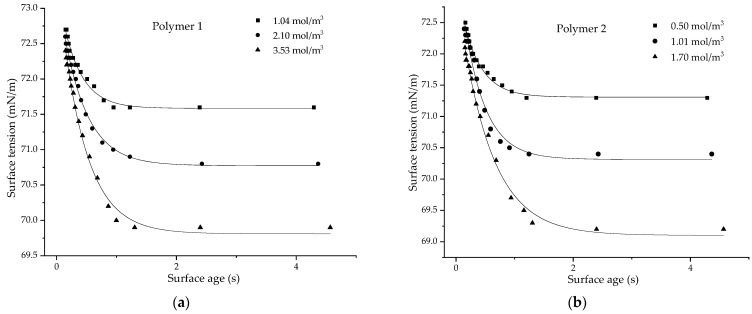
Dynamic surface tension (mN/m) at different concentrations (mol/m^3^) for CUP particles made from polymers 1–5 (**a**–**e**).

**Figure 12 polymers-14-02302-f012:**
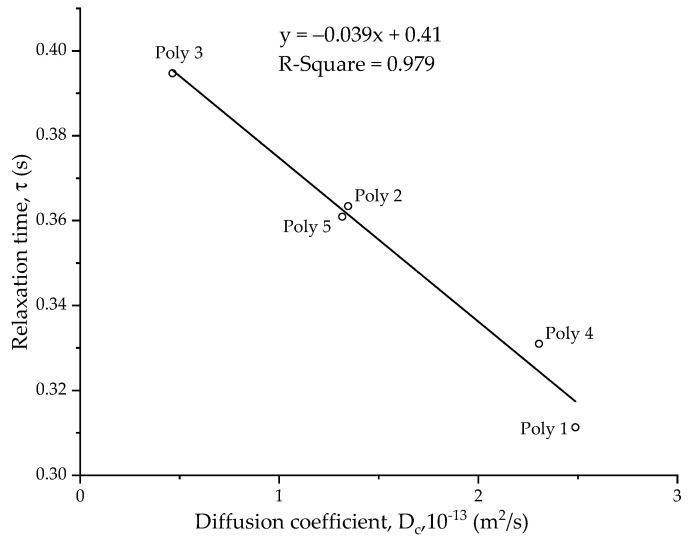
Relaxation time, τ (s), against the diffusion coefficient, D_c_ (m^2^/s), at a 1.03 mol/m^3^ concentration of CUP polymers 1–5.

**Figure 13 polymers-14-02302-f013:**
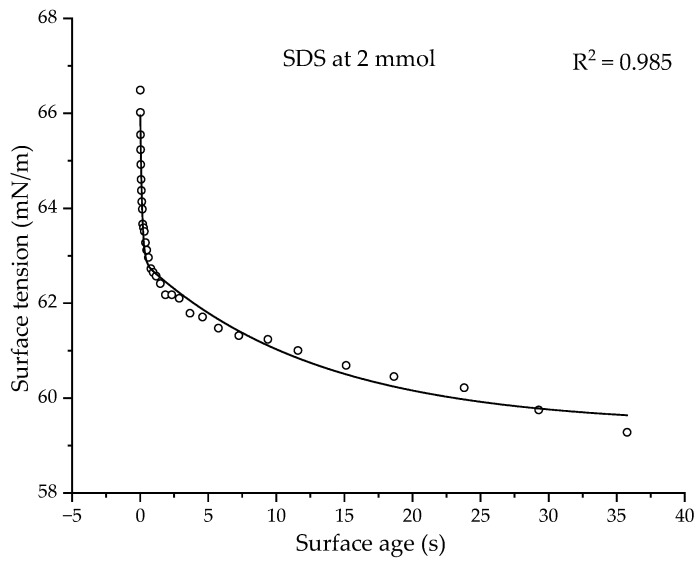
Dynamic curve of SDS at 2 mmol concentration. Reprinted (adapted) with permission from Christov, N. C., Danov, K. D., Kralchevsky, P. A., Ananthapadmanabhan, K. P., Lips, A. Maximum bubble pressure method: Universal surface age and transport mechanisms in surfactant solutions. *Langmuir*, **2006**, *22*, 7528–7542. Copyright 2019 American Chemical Society.

**Table 1 polymers-14-02302-t001:** Molar quantities of the monomers, initiator (AIBN), and chain transfer agent (1-dodecanethiol) that was used for the synthesis of polymers 1–8.

Polymer	MMA (mol)	MAA (mol)	AIBN (mol)	1-Dodecanethiol (mol)	THF (mol)
1 ^a^	0.912	0.101	7.09 × 10^−4^	3.49 × 10^−3^	2.77
2 ^a^	0.912	0.101	7.09 × 10^−4^	1.45 × 10^−3^	2.77
3 ^a^	0.912	0.101	7.09 × 10^−4^	0.76 × 10^−3^	2.77
4 ^a^	0.887	0.130	7.12 × 10^−4^	3.44 × 10^−3^	2.77
5 ^a^	0.918	0.094	7.08 × 10^−4^	1.24 × 10^−3^	2.77
6 ^a^	0.941	0.067	7.06 × 10^−4^	1.6.× 10^−3^	2.77
7	0.953	0.053	7.04 × 10^−4^	1.6 × 10^−3^	2.77
8	0.852	0.170	7.16 × 10^−4^	1.6 × 10^−3^	2.77

^a^ Data taken from Ref. [13].

**Table 2 polymers-14-02302-t002:** Acid number, densities, molecular weight, and polydispersity index of the copolymers.

Sample ID	M_W_ ^b^ (g/mol)	PDI ^c^	Monomer Ratio (MMA: MAA)	AN (mg KOH/g) ^d^	Density of Dry CUP, ρ_p_ (g/mL)
Polymer 1 ^a^	28.9 K	1.8	9:1	56.8	1.2246 ± 0.0018
Polymer 2 ^a^	59.8 K	1.7	9:1	57.0	1.2311 ± 0.0014
Polymer 3 ^a^	122.5 K	1.7	9:1	56.9	1.2342 ± 0.0018
Polymer 4 ^a^	25.4 K	2.3	6.8:1	73.2	1.2243 ± 0.0018
Polymer 5 ^a^	73.5 K	1.7	9.8:1	52.6	1.2315 ± 0.0018
Polymer 6 ^a^	49.7 K	1.8	14:1	37.7	1.2307 ± 0.0016
Polymer 7	45.4 K	1.9	18:1	29.1	1.2290 ± 0.0019
Polymer 8	50.1 K	1.6	5:1	95.8	1.2300 ± 0.0012

^a^ Data taken from Ref. [13]. ^b^ Absolute number average molecular weight from GPC. ^c^ PDI—Polydispersity index. ^d^ AN—Acid number, as measured using an ASTM D974.

**Table 3 polymers-14-02302-t003:** Measured and calculated particle size (diameter) and charge density of the CUPs.

Sample ID	d(DLS) ^b^ (nm)	d(GPC) ^c^ (nm)	Charge Density, ρ_v_, (Ions per nm^2^)
Polymer 1 ^a^	4.22	4.25	0.52
Polymer 2 ^a^	5.38	5.40	0.66
Polymer 3 ^a^	6.83	6.80	0.85
Polymer 4 ^a^	4.04	4.05	0.66
Polymer 5 ^a^	5.76	5.80	0.66
Polymer 6 ^a^	5.06	5.08	0.42
Polymer 7	4.90	4.92	0.32
Polymer 8	5.94	5.08 ^d^	1.04 ^d^, 0.83 ^e^

^a^ Data taken from Ref. [13]. ^b^ Diameters measured by a dynamic light scattering (DLS) instrument. ^c^ Diameters calculated from the average molecular weight, measured using gel permeation chromatography (GPC) using Equation (1). ^d^ Assuming a sphere conformation. ^e^ Assuming a dumbbell conformation.

**Table 4 polymers-14-02302-t004:** Comparison of the surface tension of CUPs surfactants, sodium chloride and sodium carboxylates.

	Concentration c/c* ^b^, mol/L	Surface Tension ^a^, γ, mN/m	Δγ ^d^	Δγ/Δc (Δγ/Δc*) mN·m^2^/mol (×10^3^)
Water	0	72.2	0.0	0
CUPs (Polymer 3)	0.001/0.0155 ^b^	68.0	4.2	−4200 (−271)
CUPs (Polymer 2)	0.001/0.0114 ^b^	70.3	1.9	−1900 (−166)
CUPs (Polymer 7)	0.001/0.0067 ^b^	71.9	0.3	−300 (−45)
QUAT CUPs ^c^	0.001	68.7	3.5	−3500
Sulfonate CUPs ^c^	0.001	65.6	6.6	−6600
SDS ^c^	0.001	65.0	7.2	−7200
Sodium Chloride	0.35	73.9	−1.7	4.86
Sodium Formate ^c^	1	73.2	−1.0	1
Sodium Acetate ^c^	1	70.2	2.0	−2
Sodium Benzoate ^c^	0.26	68.2	4.0	−15.38
Sodium Laurate ^c^	0.001	63.6	8.6	−8600
Sodium Heptanoate	0.005	70.5	1.7	−340
Sodium Octanoate	0.005	65.5	6.7	−1340

^a^ The surface tension values are below the CMC. ^b^ The concentrations c* for CUP polymers 2, 3 and 7 were calculated using Equation (7). ^c^ Data taken from Refs. [19,21,40]. ^d^ Δγ = γ_water_ − γ_CUP_.

**Table 5 polymers-14-02302-t005:** Relaxation time (*τ_k_*) for the CUP particles of polymer 1–5 at different concentrations.

Sample ID	Concentration, mol/m^3^	γ_e_, mN/m	τ, s	A	R^2^
Polymer 1	1.04	71.59	0.311	1.77	0.983
	2.10	70.77	0.401	2.57	0.993
	3.53	69.81	0.426	3.69	0.994
Polymer 2	0.50	71.31	0.315	1.84	0.984
	1.01	70.31	0.363	3.27	0.981
	1.70	69.10	0.543	3.99	0.993
Polymer 3	0.24	71.25	0.264	2.64	0.995
	0.49	69.83	0.360	4.22	0.994
	1.02	68.08	0.395	5.59	0.995
Polymer 4	1.05	70.55	0.331	2.63	0.994
	2.39	68.05	0.374	4.22	0.997
	4.03	64.99	0.406	5.59	0.995
Polymer 5	0.41	71.50	0.330	2.01	0.985
	1.02	70.66	0.361	3.28	0.998
	1.45	69.62	0.424	3.76	0.998

**Table 6 polymers-14-02302-t006:** Particle size, charge density, relaxation time and diffusion coefficient of CUP polymers 1–5, measured at the average concentration of 1.03 ± 0.02 mol/m^3^.

Sample ID	Particle Size, nm	Charge Density, Ions/nm^2^	Relaxation Time, τ	A	Diffusion Coefficient 10^−13^ m^2^/s
Polymer 1	4.22	0.52	0.311	1.77	2.49
Polymer 2	5.38	0.66	0.363	3.27	1.35
Polymer 3	6.28	0.84	0.395	5.59	0.46
Polymer 4	4.04	0.66	0.331	2.63	2.30
Polymer 5	5.50	0.66	0.361	3.28	1.32

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
