# Peer review of "Equilibrium and Dynamic Surface Tension Behavior in Colloidal Unimolecular Polymers (CUP)"

_polymers, 2022, doi:10.3390/polym14112302_

Round 1

Reviewer 1 Report

Governing the surface tension of liquids is an important task of variety of applications. This defines the soundness of the present study offering the ways to change the surface tension of different liquids in desired manner by CUP particles. It is shown the particle size and charge density​ are key factors determining the air/liquid/particle interface property. The model developed is quite accurate since it adequately takes into account many relevant parameters. As such, the demonstrated correspondence between the experiment and the model is convincing.
The strength of the work: Combination of state of the art sample preparation techniques and complementary characterization tools and the modeling enabling quite convincing data and interpretation.
The weakness: The work would gain if particle size distribution data were added by direct technique, for instance, scanning electron microscopy. Also direct application of DLS to nanosize particles looks somewhat doubtful, since the wavelength of light is larger than the particle size (3-9 nm).
In general, the work is scientifically sound, well arranged and conducted, accurately presented, referencing is comprehensive and up to date. In my view, the manuscript is suitable for publication in Polymers in its present form.

--

Reviewer 2 Report

The paper is interesting in its field. However, changes are recommended before further consideration. The main issue resides in the fact that the authors appear to have synthesized 2 materials for this work, but the section of results and discussion takes results from a previous paper on other 5 similar polymers. Later on, results are presented for all polymers so it is not clear of they were from previous work of the measurements have been performed on all polymers? Moreover, by the end of the paper, the authors report results only on the polymers that were synthesized and reported in the previous work and no results on the new ones are included. Another issue is that the presentation lack clarity: data is presented before the formulas, discussion is presented in a different part then the related figure/table.. It makes the paper difficult to follow. Some of examples on these issues are appended below, but the authors are advised to revise the whole manuscript:

  • Abstract: please include also values in terms of your findings/novelty
  • Experimental: although the relevant polymerization information for polymers 1-6 is reported elsewhere, the authors later discuss on the properties of all polymers so, for the sake of better understanding and straightforwardness to the reader, such information should be presented along with polymers 7 and 8 in table 1 in supplementary file.
  • Results and discussion: section 3.1- this a simple presentation of polymers 7 & 8 synthesized in the present work in comparison with other 5 polymers reported in an earlier work. Only experiments and results from the present work should be reported, while the rest could be included in supplementary file. There is no information on the characteristics such as acid number, density, molecular weight, and polydispersity index of the copolymers – how were these measured? There is no discussion!
  • Results and discussion: section 3.2 - Information from section 3.2 should be moved to experimental one. Data reported in table 3: please add reference support for considering sphere and dumbbell conformations/ related equations. Again, only 2 polymers should be reported here since the others were presented in other paper.
  • Results and discussion: section 3.3- table 4- why there is no result on polymer 8 (one of the 2 new polymers reported in this work)? For concentration values reported in Table 4 it is mentioned that eq. 7 was used but there is no such eq. presented before. It appears that discussions are not presented along with the tables/figures, the authors should improve the correlation and clarity in presentation.
  • Results and discussion: fig 11&12/ table 5&6- why the new polymers that were synthesized in this work are not presented?

Reviewer 3 Report

In their manuscript entitled "Equilibrium and Dynamic Surface Tension Behavior in Colloidal Unimolecular Polymers (CUP)", the authors describe the synthesis (briefly) and characterization of the surface tension of aqueous dispersions of nanoparticles of methacrylate-methacrylic acid polymer blends. The study is very comprehensive and comprises of the discussion of the experimental results for the equilibrium and dynamic surface tension by the bubble pressure method and of the model description of the observed dependences of the surface energy on particle properties (size, molecular weight, charge) and observation time.

The manuscript is well structured and well written. Only minor typos require some consideration when proof-reading the manuscript by the authors:

Abstract, line 20: …CUP particles…

Page 6, line 224: …higher surface activity.

Page 6, line 225: …values of sodium acetate…

Table 4: concentration unit mol/l; add unit for charge of surface tension with concentration

Page 9, line 284: …contribution of thioglycolic…

Page 9, line 289: …couple of days…

Page 13, lines 391-392: …related to charge density.

Figure 7: delete (N_charge) in the first bracket

Page 16, line 454: 3.7. Dynamic surface tension behavior

Page 20, line 533: …CUP particles provide a…

The affiliation is the same for all authors. There is no obvious need to separare these in (1) and (2).

Figure 2 and 3 show the same results (surface tension vs. concentration) with different units of concentration (molar and weight concentration). These figures may be combined (e.g., by using two x-axes).

Page 10, line 341: The authors state that the particles do not experience any charge forces extered on them from the nitrogen-water interface. Alike polymer-water and oil-water interfaces, gas-water interfaces are supposed to accumulate an interfacial charge (e.g., the accumulation of water ions). I wonder if this needs some consideration by the authors when describing the electrostatic forces applied on the particles.

Round 2

Reviewer 2 Report

The review comments were addressed.